# Major role of iron uptake systems in the intrinsic extra-intestinal virulence of the genus *Escherichia* revealed by a genome-wide association study

Marco Galardini[1¤]*, Olivier Clermont[2], Alexandra Baron[2], Bede Busby[3], Sara Dion[2], Sören Schubert[4], Pedro Beltrao[1], Erick Denamur[2,5]*

**1** EMBL-EBI, Wellcome Genome Campus, Cambridge, United Kingdom, **2** Université de Paris, IAME, UMR1137, INSERM, Paris, France, **3** Genome Biology Unit, EMBL, Heidelberg, Germany, **4** Max von Pettenkofer Institute of Hygiene and Medical Microbiology, Faculty of Medicine, LMU Munich, Germany, **5** AP-HP, Laboratoire de Génétique Moléculaire, Hôpital Bichat, Paris, France

¤ Current address: Biological Design Center, Boston University, Boston, MA, United States of America
* mgala@bu.edu (MG); erick.denamur@inserm.fr (ED)

**Data Availability Statement:** All input data and code used to run the analysis and generate the

## Abstract

The genus *Escherichia* is composed of several species and cryptic clades, including *E. coli*, which behaves as a vertebrate gut commensal, but also as an opportunistic pathogen involved in both diarrheic and extra-intestinal diseases. To characterize the genetic determinants of extra-intestinal virulence within the genus, we carried out an unbiased genome-wide association study (GWAS) on 370 commensal, pathogenic and environmental strains representative of the *Escherichia* genus phylogenetic diversity and including *E. albertii* (n = 7), *E. fergusonii* (n = 5), *Escherichia* clades (n = 32) and *E. coli* (n = 326), tested in a mouse model of sepsis. We found that the presence of the high-pathogenicity island (HPI), a ~35 kbp gene island encoding the yersiniabactin siderophore, is highly associated with death in mice, surpassing other associated genetic factors also related to iron uptake, such as the aerobactin and the *sitABCD* operons. We confirmed the association *in vivo* by deleting key genes of the HPI in *E. coli* strains in two phylogenetic backgrounds. We then searched for correlations between virulence, iron capture systems and *in vitro* growth in a subset of *E. coli* strains (N = 186) previously phenotyped across growth conditions, including antibiotics and other chemical and physical stressors. We found that virulence and iron capture systems are positively correlated with growth in the presence of numerous antibiotics, probably due to co-selection of virulence and resistance. We also found negative correlations between virulence, iron uptake systems and growth in the presence of specific antibiotics (*i.e.* cefsulodin and tobramycin), which hints at potential "collateral sensitivities" associated with intrinsic virulence. This study points to the major role of iron capture systems in the extra-intestinal virulence of the genus *Escherichia*.

plots is available online at https://github.com/
mgalardini/2018_ecoli_pathogenicity.

**Funding:** This work was partially supported by the
"Fondation pour la Recherche Médicale" (Equipe
FRM 2016, grant number DEQ20161136698). The
funders had no role in study design, data collection
and analysis, decision to publish, or preparation of
the manuscript.

**Competing interests:** The authors have declared
that no competing interests exist.

## Author summary

Bacterial isolates belonging to the genus *Escherichia* can be human commensals but also
opportunistic pathogens, with the ability to cause extra-intestinal infection. There is there-
fore the need to identify the genetic elements that favour extra-intestinal virulence, so that
virulent bacterial isolates can be identified through genome analysis and potential treat-
ment strategies be developed. To reduce the influence of host variability on virulence, we
have used a mouse model of sepsis to characterize the virulence of 370 strains belonging
to the genus *Escherichia*, for which whole genome sequences were also available. We have
used a statistical approach called Genome-Wide Association Study (GWAS) to show how
the presence of genes that encode for iron scavenging are significantly associated with the
propensity of a bacterial isolate to cause extra-intestinal infections. Taking advantage of
previously generated growth data on a subset of the strains and its correlation to virulence
we generated hypothesis on the relationship between iron scavenging and growth in the
presence of various antimicrobials, which could have implications for developing new
treatment strategies.

## Introduction

Members of the *Escherichia* genus are both commensals of vertebrates [1] and opportunistic
pathogens [2] involved in a wide range of intestinal and extra-intestinal infections. Apart from
the *E. coli* species, the genus is composed of the cryptic *Escherichia* clades, and the *E. fergusonii*
and *E. albertii* species. The latter taxa are rarely isolated in humans but are more frequently
found in the environment and avian species where they can cause intestinal infections [3–5].
In humans, extra-intestinal infections represent a considerable burden [6], with bloodstream
infections (bacteraemia) being the most severe with a high attributable mortality of between
10–30% [7–10]. The regular increase over the last 20 years of *E. coli* bloodstream incidence
[11] and antibiotic resistance [12] is particularly worrisome. The factors associated with high
mortality are mainly linked to host conditions such as age, the presence of underlying diseases
and to the portal of entry, with the urinary origin being more protective. These factors out-
weigh those directly attributable to the bacterial agent [7–9,13].

Nevertheless, the use of animal models has shown a great variability in the intrinsic extra-
intestinal virulence potential of natural *Escherichia* isolates. In a mouse model of sepsis where
bacteria are inoculated subcutaneously, it has been clearly shown that the intrinsic virulence
quantified by the number of animal deaths over the number of inoculated animals for a given
strain is dependant on the number of virulence factors such as adhesins, toxins, protectins and
iron capture systems [14–19]. One of the most relevant virulence factors is the so-called high-
pathogenicity island (HPI), a 36 to 43 kb region encoding the siderophore yersiniabactin, a
major bacterial iron uptake system [20], which has also been shown to reduce the efficacy of
innate immune cells to cause oxidative stress [21]. The deletion of the HPI results in a decrease
in the intrinsic virulence in the mouse model in a strain-dependent manner [16,18,22], indi-
cating complex interactions between the genetic background of each strain and the HPI.

The limitation of these gene inactivation studies is that they target specific candidate genes
and cannot be performed in a large number of strains. Recently, the development of new
approaches in bacterial genome-wide association studies (GWAS) [23–26] allows searching in
an unbiased manner for genotypes associated with specific phenotypes such as drug resistance
or virulence in numerous strains. In this context, we conducted a GWAS in 370 commensal
and pathogenic strains of *E. coli*, and related *Escherichia* clades, as well as *E. fergusonii* and *E.*

*albertii*, representing the genus phylogenetic diversity, to search for traits associated with virulence in the mouse model of sepsis [27]. Most of the strains were isolated from a human host and are divided between commensals and extra-intestinal pathogens. Most importantly, many (N = 186) of these strains have been recently phenotyped across hundreds of growth conditions, including antibiotics and other chemical and physical stressors [28]. This data could then be used to find phenotype associations with virulence and to generate hypotheses on the function of genetic variants associated with the extra-intestinal virulence phenotype and their role for growth in those conditions.

## Results

### GWAS identifies the high-pathogenicity island as the strongest driver of the extra-intestinal virulence phenotype

We studied a 326 strain collection representative of the *E. coli* phylogenetic diversity, with strains belonging to phylogroups A (N = 72), B1 (N = 41), B2 (N = 111), C (N = 36), D (N = 20), E (N = 19), F (N = 12) and G (N = 15). To have a broader phylogenetic representation, which could increase statistical power [24,29], we also included strains from *Escherichia* clades I to V (N = 32) and the species *E. albertii* (N = 7) and *E. fergusonii* (N = 5) [30]. These strains encompass 170 commensal strains and 187 strains isolated in various extra-intestinal infections, mainly urinary tract infections and bacteraemia [7,14,31–37]. The isolation host is predominantly humans (N = 291), followed by animals (N = 72) and isolates from environmental sources (N = 6). To avoid any bias linked to host conditions, we assessed the strain virulence as its intrinsic extra-intestinal pathogenic potential using a well-calibrated mouse model of sepsis [14,27], expressed as the number of killed mice over the 10 inoculated per strain. In accordance with previous data [14,17,27,38,39], phylogroup B2 is the most associated with the virulence phenotype ($2E^{-9}$ Wald test p-value, Fig 1A, S1 Table).

We used a bacterial GWAS method to associate unitigs—which are nodes in a colored de Bruijn graph representing a contiguous DNA sequence shared by one or more samples—to the virulence phenotype, allowing us to simultaneously test the contribution of core and accessory genome variation to pathogenicity [25]. It is generally understood that such methods require large sample sizes and phylogenetic diversity to have sufficient power, due to the need to observe multiple independent acquisitions of causal variants across clades and distinguish them from lineage defining variants; the appropriate sample size is also a function of the penetrance of the causal variants [24,29]. We ran simulations with an unrelated set of complete *E. coli* genomes and verified that our sample size was appropriate for variants with high penetrance and intermediate frequency (i.e. odds ratio above 5 and minor allele frequency > 0.1, S1 Fig, Methods). We reasoned that some of the genetic determinants of virulence are likely to have a relatively high penetrance due to the selective advantage they might confer in opening up a new niche [40,41], and that the strains used were phylogenetically diverse, enough to reach sufficient statistical power.

We uncovered a statistically significant association between 5,214 unitigs and the virulence phenotype, which were mapped back to 81 genes across the strains' pangenome (Fig 1B, S2 Table, Methods). We carried out a gene ontology (GO) term enrichment analysis on the 81 genes, and found that 7 terms were significantly enriched (FDR-corrected p-value < 0.05, S3 Table); among those 6 were related to iron homeostasis (such as GO:0030091, "response to iron ion"), and one to protein repair (GO:0030091). To understand whether the presence of these 81 genes is directly associated with virulence or if it is due to genetic variants such as SNPs we performed a separate association analysis using genes' presence absence patterns. This showed that most genes have an odds ratio that far exceeds the required threshold we

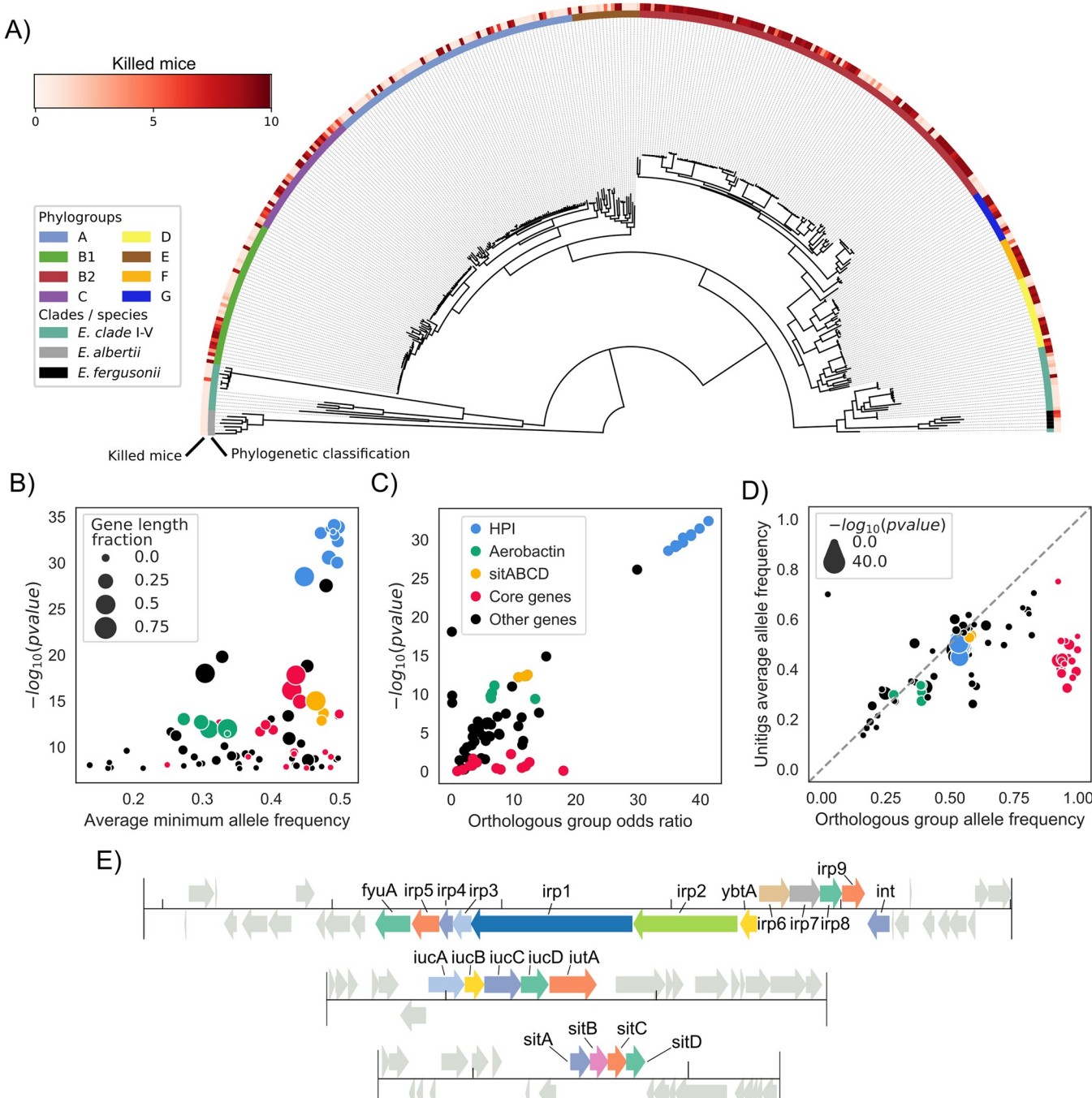

**Fig 1. The HPI is strongly associated with the extra-intestinal virulence phenotype assessed in the mouse sepsis assay.** A) Core genome phylogenetic tree of the *Escherichia* strains used in this study rooted on *E. albertii* strains. Outer ring reports virulence as the number of killed mice over the 10 inoculated per strain, inner ring the phylogroup, clade or species each strain belongs to. B) Results of the unitigs association analysis: for each gene the minimum association p-value and average minimum allele frequency (MAF) across all mapped unitigs is reported. The gene length fraction is computed by dividing the total length of mapped unitigs by the length of the gene. The color of each gene follows the same key as panel C. C) Results of the gene presence/absence association analysis; only those genes with at least one associated unitig mapped to them are represented. D) Scatterplot of gene frequency versus frequency of associated unitigs; points on the diagonal indicate hits where the association is most likely due to a gene's presence/absence pattern rather than a SNP. The color of each gene follows the same key as panel C. E) The structure of the HPI and of the aerobactin and *sitABCD* operons in strain IAI39; all associated genes are highlighted.

estimated from simulations, as well as low association p-value (Fig 1C). Furthermore, 48 out of 81 genes with at least one associated unitig mapped to them have a frequency across strains that is highly correlated with that of the associated unitigs (Fig 1D), indicating that it's the presence/absence pattern of those genes to be associated with virulence and not other kinds of genetic variants such as SNPs mapping to those genes.

Genes belonging to the HPI had the lowest association p-value by far ($<1E^{-28}$); the presence of genes belonging to two additional operons encoding for bacterial siderophores (aerobactin [42] and *sitABCD* [43]) was also found to be associated with virulence (Fig 1E). We found that the HPI structure was highly conserved across the genomes that encode it (S2 Fig). We also observed that the distribution of a collection of some known virulence factors [44] didn't match the virulence phenotype as closely as the HPI or the aerobactin and *sitABCD* operons, or had unitigs passing the association threshold (p-value $> 2.16E^{-08}$, gene presence/absence patterns shown in S4 Fig), suggesting how iron scavenging is an important factor in determining virulence.

Among the remaining 33 genes with associated unitigs out of 81 total, 18 have a high frequency in the pangenome ($> 0.9$) and a low gene length fraction (i.e. the associated unitigs cover only a fraction of the gene, $< 50\%$, Fig 1B), indicating that the presence of genetic variants such as SNPs present in core genes is associated to the virulence phenotype. We found that the core genes with the lowest association p-values were: *zinT* (p-value $1E^{-16}$), encoding a zinc and cadmium binding protein [45], *mtfA* (p-value $1E^{-14}$), encoding a protein involved in the regulation of carbohydrate metabolism [46], *shiA* (p-value $1E^{-14}$), encoding a transporter of shikimate, a compound involved in siderophore synthesis [47,48], *hprR* and *hprS* (p-value $1E^{-13}$ and $1E^{-9}$, respectively), encoding a two-component regulatory systems involved in the response to hydrogen peroxide [49] and *msrPQ* (p-value $1E^{-12}$ for both genes) an operon encoding enzymes involved in repairing periplasmic proteins under oxidative stress [50]. Most of these core genome hits (14 over 18 total) are encoded in the region surrounding the HPI (S3 Fig), which might imply that these hits are correlated with the presence of the HPI and not causally linked with extra-intestinal virulence. The remaining four core genome hits include *rspB* (p-value $1E^{-8}$), encoding a starvation sensing protein, and *torD* (p-value $1E^{-8}$), part of the *torCAD* operon involved in anaerobic respiration with trimethylamine-N-oxide (TMAO) as an electron acceptor [51,52].

### Gene knockout experiments validate the role of the HPI in the extra-intestinal phenotype

Previous studies on the role of the HPI in experimental virulence gave contrasting results according to the strains' genetic background [18]. Among B2 phylogroup strains, HPI deletion in the 536 strain (ST127; ST: sequence type) did not have any effect in the mouse model of sepsis [53] whereas this deletion in the NU14 strain (ST95) dramatically attenuated virulence [18]. Two strains from the present study belonging to B2 phylogroup/ST141 (IAI51 and IAI52) deleted in the longest gene of the HPI (*irp1*) have attenuated virulence in the same mouse model [22]. Deletion of the second longest gene of the HPI (*irp2*) in a strain (A1749) belonging to phylogroup D (ST69) also showed attenuated virulence in the same sepsis model [54]. We further documented the role of the HPI in extraintestinal virulence constructing *irp2* deletion gene mutants in two additional strains of phylogroup D (NILS46, ST69) and A (NILS9, ST10) completing the panel of sequence types frequently involved in human bacteraemia [55]. We first verified that the wild-type strains strongly produced yersiniabactin, whereas both *irp2* mutants did not (Fig 2A). We then tested them in the mouse sepsis model and saw an increase in survival for both mutated strains (log-rank test p-value 0.02 and $< 0.0001$ or

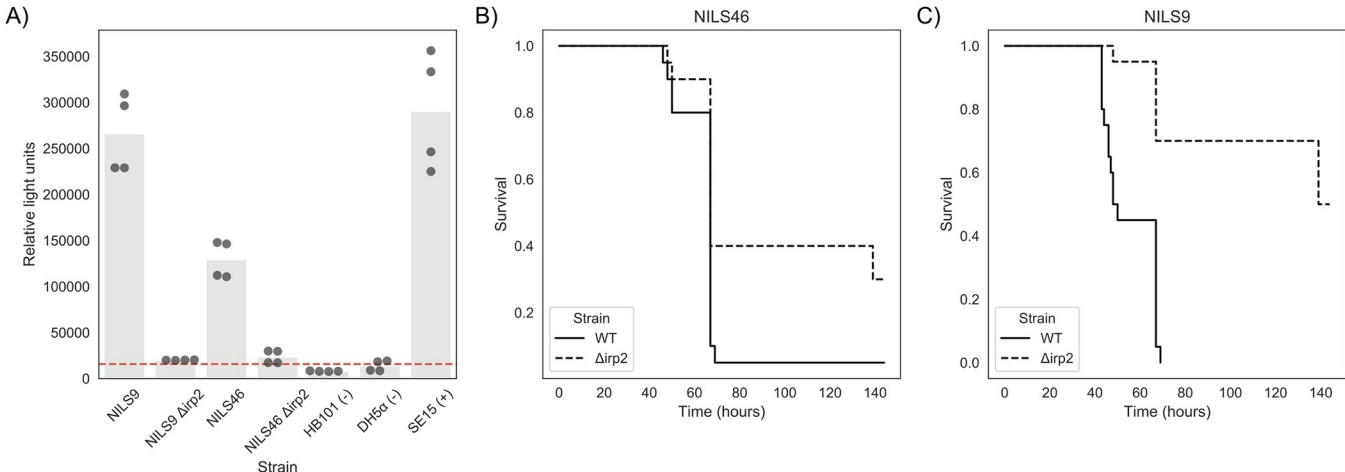

**Fig 2. Phenotypic consequences of HPI deletion.** A) Deletion of HPI leads to a decrease in production of yersiniabactin. Production of yersiniabactin is measured using a luciferase-based reporter (Methods). Strains marked with a "-" and "+" sign indicate a negative and positive control, respectively. The red dashed line indicates an arbitrary threshold for yersiniabactin production, derived from the average signal recorded from the negative controls plus two standard deviations. B-C) Deletion of HPI leads to an increase in survival after infection. Survival curves for wild-type strains and the corresponding *irp2* deletion mutant, built after infection of 20 mice for each strain. B) Survival curve for strain NILS46. C) Survival curve for strain NILS9.

strain NILS46 and NILS9, respectively, Fig 2B and 2C, S4 Table) with no significant difference between the survival profiles for the two mutants (log-rank test p-value > 0.1). We therefore bring additional experimental evidence of the role of the HPI in extra-intestinal virulence. A much larger sample size would be required to evidence a dependency on genetic background for the relationship between HPI and virulence. Nevertheless, we have validated the causal link between the HPI and the virulence phenotype *in vivo* which demonstrates the power and accuracy of bacterial GWAS.

## High-throughput phenotypic data sheds light on HPI and other iron capture systems functions

The main function encoded by the HPI cassette is iron scavenging through the expression of the siderophore yersiniabactin [22], which has been previously validated in *E. coli* through knockout experiments [18]. The aerobactin operon also encodes an iron chelator [42], while the *sitABCD* operon encodes a $Mn^{2+}/Fe^{2+}$ ion transporter [43]. In order to investigate other putative functions of these operons and their relationship with virulence, we leveraged a previously-generated high-throughput phenotypic screening in an *E. coli* strain panel that largely overlaps with the strains used here (186 strains over 370 analyzed in this study) [28]. We observed a relatively strong correlation (Pearson's correlation p-value < 1E-4) between growth profiles in certain in vitro conditions and both virulence and presence of the HPI, aerobactin and *sitABCD* operons (Fig 3A–3D, S5 Table).

As expected, we found a positive correlation between growth on the iron-sequestering agent pentetic acid [56] and both virulence and HPI/aerobactin/*sitABCD* presence (Pearson's r: 0.36, 0.48, 0.23 and 0.39, respectively). We also found that growth in the presence of bipyridyl, an iron chelator, was positively correlated with the presence of aerobactin (exact condition: bipyridyl + tobramycin, Pearson's r: 0.30). We similarly observed a positive correlation between growth with pyocyanin, a redox-active phenazine compound being able to reduce $Fe^{3+}$ to $Fe^{2+}$ [57], and both HPI/aerobactin/*sitABCD* presence (Pearson's r: 0.35, 0.28, 0.26 and 0.27 respectively). All these mentioned growth conditions have a correlation sign that agrees

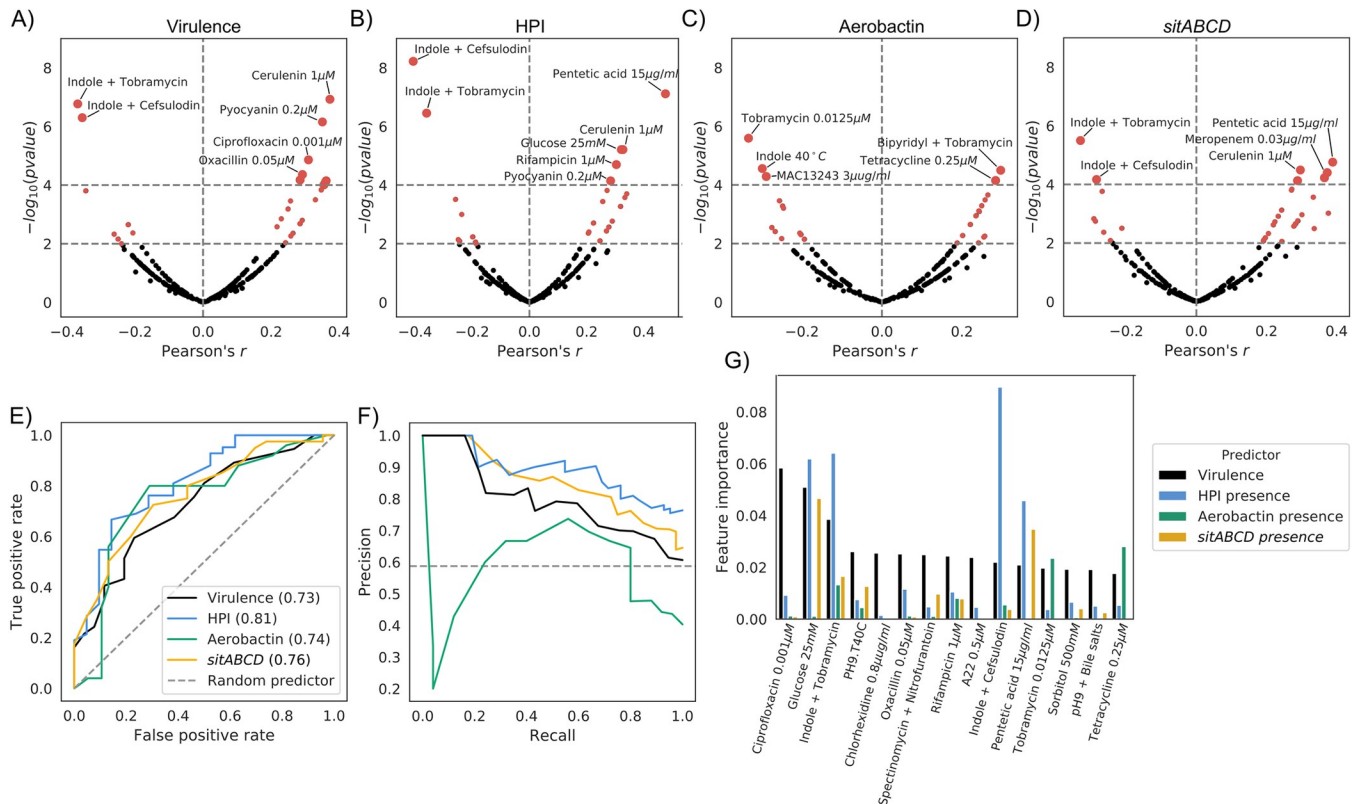

**Fig 3. Growth profiles can predict virulence and presence of virulence factors.** A-D) Volcano plots for the correlation between the strains' growth profiles and: A) virulence levels, B) presence of the HPI, C) presence of aerobactin, and D) presence of *sitABCD*. E-F) Use of the strains' growth profiles to build a predictor of virulence levels and presence of the three iron uptake systems. E) Receiver operating characteristic (ROC) curves and F) Precision-Recall curve for the four tested predictors. G) Feature importance for the predictors, showing the top 15 conditions contributing to the virulence level predictor.

with the iron scavenging function of the three gene clusters and their importance for virulence.

Interestingly, we also found similarly strong positive correlations between virulence and presence of iron capture systems with growth on sub-inhibitory concentrations of several antimicrobial agents, such as rifampicin, ciprofloxacin, tetracycline and ß-lactams such as amoxicillin, oxacillin, meropenem, cerulenin and colicin. These correlations might be due to the presence/absence of acquired resistance alleles and/or genes that are strongly associated with pathogenic strains, or might point to the role of iron homeostasis in intrinsic resistance to antibiotics [53]. To investigate these two hypotheses, we focused on tetracycline resistance, a common occurrence in the genus [34,55,58], and for which resistance genes can be easily found through sequence homology (Methods). We measured the correlation between the presence of tetracycline resistance genes, found in 26.8% of the strains, and virulence (Pearson's r: 0.16), as well with the presence of either of the three iron capture systems (Pearson's r: 0.21, 0.33 and 0.24 for HPI, aerobactin and *sitABCD*, respectively), which we found to be comparable in terms of sign and magnitude with the direct correlation between growth on sub-inhibitory concentration of tetracycline and the presence of resistance genes (Pearson's r: 0.4). These correlations between virulence, iron capture systems and growth in the presence of tetracycline are however greatly reduced (Pearson's r < 0.1) when correcting for the presence of tetracycline resistance genes using partial correlation. This suggests that there might not be a direct relationship between virulence, the GWAS hits and growth in the presence of tetracycline.

On the other hand we found that growth in presence of indole at 2 mM either in association with sub-inhibitory concentrations of cefsulodin and tobramycin, or alone at 40°C was negatively correlated with both virulence and HPI/aerobactin/*sitABCD* presence. Similar negative correlation was observed with aerobactin presence and the MAC13243 compound that increases outer membrane permeability [59]. This indicates that there might be a trade-off between growth in these conditions and virulence, *i.e.* virulent strains are less fit when growing in the presence of these compounds.

Given the relatively large number of conditions correlated with both virulence and presence of iron uptake systems, we tested whether these features could be predicted from growth profiles. We used the commonly-used random forests machine learning algorithm with appropriate partitioning of input data into training and test sets to tune hyperparameters and reduce overfitting (Methods). We trained and tested four classifiers for virulence and presence of the HPI, aerobactin and *sitABCD* operons with high predictive power, with the exception of aerobactin, which performed slightly worse, although still better than an empirical random (Fig 3E and 3F, S5 Fig and Methods). We noted that prediction of the gene clusters presence performs slightly better than virulence, possibly reflecting the complex nature of the latter phenotype. As expected, we found that conditions with relatively high correlation with each feature have a higher weight across classifiers (Fig 3G, S6 Table), which suggests that a subset of phenotypic tests might be sufficient to classify pathogenic strains. These results show how phenotypic data can be used to generate hypotheses for the function of virulence factors.

## Discussion

With the steady decline in the price of genomic sequencing and the increasing availability of molecular and phenotypic data for bacterial isolates, it has finally become possible to use statistical genomics approaches such as GWAS to uncover the genetic determinants of relevant phenotypes. Such approaches have the advantage of being unbiased, and can then be used to confirm previous targeted findings and potentially uncover new factors, given sufficient statistical power. The accumulation of other molecular and phenotypic data can on the other hand uncover variables correlated with phenotype, which can be used to generate testable hypotheses on the function of genomic hits and their role for growth in those correlated conditions. Given the rise of both *E. coli* extra-intestinal infections and antimicrobial resistance, we reasoned that the intrinsic virulence assessed in a calibrated mouse model of sepsis [14,27] is a phenotype worth exploring with such an unbiased approach.

Our work points to the fundamental role of iron scavenging in the extra-intestinal virulence phenotype in the genus *Escherichia* [60]. In fact, we found that 6 over the 7 GO terms significantly enriched were related to iron homeostasis. We were able to confirm earlier reports on the importance of the presence of the HPI in extra-intestinal virulence [18–20,22,54,61], which showed the strongest signal in both the unitigs and accessory genome association analysis, and whose importance was validated *in vitro* and in an *in vivo* model of virulence. The distribution of the HPI within the species resulting from multiple horizontal gene transfers via homologous recombination [62] has probably facilitated its identification using GWAS, since these methods favor the discovery of elements that are independently acquired across clades. We associated additional genetic factors to intrinsic virulence, such as the presence of the aerobactin and *sitABCD* operons, both related to iron scavenging together with the HPI. We also found mutations in core genes such as *hprRS* and *msrPQ* to be associated with virulence, whose role in response to oxidative stress and protein repair is compelling, although their association to virulence might be due to their physical proximity to the HPI. Thus, genetic variants in these genes could be associated with virulence through hitchhiking [62]. Hits in other core genes

such as *rspB*, related to starvation sensing are similarly compelling. *rspB* is part of an operon with *rspA*, a gene encoding a protein involved in the degradation of homoserine lactone that signals starvation [63]. Further genetic and molecular characterization might elucidate the role of these core genes' variants in extra-intestinal virulence. Additional factors might have been overlooked by this analysis, due to the relatively small sample size; we however estimate that those putative additional factors might have a relatively low penetrance, based on our simulations in an independent dataset. As sequencing of bacterial isolates is becoming more common in clinical settings [64–66], we expect to be able to uncover these additional genetic factors in future studies.

The association between both the intrinsic virulence phenotype and the presence of the virulence factors—such as the HPI—and previously collected growth data allowed us to generate hypotheses on mechanism of pathogenesis and putative additional functions of these factors. In particular we observed a strong correlation between growth on various antimicrobial agents and both virulence and HPI/aerobactin/*sitABCD* presence, which may be the result of the acquisition of both resistance genes/alleles and iron capture genes in these isolates, as exemplified for tetracycline resistance genes. This could be explained by a greater exposure to antibiotics and subsequent selection of resistance in clinical virulent strains, leading to the positive correlation we have observed. As such there might not be a causal relationship between increased iron uptake and antimicrobial resistance, but rather the two phenotypes coincide because of their selective advantage in the context of extra-intestinal pathogenesis.

The negative correlation between virulence and iron capture systems and growth profiles in the presence of 2 mM indole associated with stress conditions such as sub-lethal doses of specific antibiotics (cefsulodin and tobramycin) or high temperature but not indole alone, points however to the possible deleterious role of iron in such conditions. In *E. coli* cells grown in lysogeny broth in planktonic [67] or biofilm [68] conditions, sub-lethal concentrations of numerous antibiotics (ampicillin, trimethoprim, nalidixic acid, rifampicin, kanamycin and streptomycin) increase the endogenous production of indole to 1.5–6 mM. The production of indole is dependent on the amount of exogenous tryptophan, and it is conceivable that this range of indole concentrations obtained *in vitro* can be produced in the mammalian host [69] Indole is toxic for the cells above 3–5 mM, as it induces the production of reactive oxygen species and prevents cell division by modulating membrane potential [70,71]. A vicious circle is rapidly established as antibiotics increase the production of indole [67], which in turn destabilises the membrane [70,71], further increasing the penetration of the antibiotics. The toxicity of indole has been shown to be partly iron mediated due to the Fenton reaction, the deletion of TonB, an iron transporter, increasing resistance to the antibiotic [72]. Sub-lethal doses of tobramycin leads to an increase of reactive oxygen species in the bacterial cell in relation to intra-cellular iron and the Fenton reaction [73]. Thus, cells with increased import of extracellular iron might be more sensitive to sub-lethal doses of specific antibiotics, suggesting a potential "collateral sensitivity" related to both intrinsic virulence and the presence of the iron uptake systems. The expression "collateral sensitivity" is normally used to refer to selection for one antibiotic resistance resulting in increased sensitivity to a second antibiotic [74]. Here we propose to extend its meaning to include the negative correlation observed in this study; that is, the trade-off between the benefits brought by iron scavenging systems in one trait (virulence) being linked to detrimental changes in other traits (antibiotic sensitivity). Altogether, these data bring new light on the "liaisons dangereuses" between iron and antibiotics that could potentially be targeted [75]. More generally, they show that the presence of iron capturing systems can be either advantageous or disadvantageous, depending on the growth conditions. Further studies will however be needed to confirm this proposed "collateral sensitivity" and its molecular mechanism.

In conclusion, we showed the power of bacterial GWAS to identify major virulence determinants in bacteria. Within the *Escherichia* genus, iron capture systems seem to be the main predictors of the intrinsic extra-intestinal virulence, at least according to the mouse model of sepsis used here. Furthermore, this analysis demonstrates how a data-centric approach can increase our knowledge of complex bacterial phenotypes and guide future empirical work on gene function and its relationship to intrinsic virulence.

## Materials and methods

### Strains used

The full list of the 370 strains used in the association analysis, together with their main characteristics is reported in S1 Table. These strains belong to various published collections: ECOR (N = 71) [31], IAI (n = 81) [14], NILS (N = 82) [33], Septicoli (N = 39) [10], ROAR (N = 30) [34], Guyana (N = 12) [32], Coliville (N = 8) [35], FN (N = 6) [36], COLIRED (N = 3) [37], COLIBAFI (N = 2) [7], correspond to archetypal strains (N = 7) or are miscellaneous strains from our personal collections (N = 29). The isolation host is predominantly humans (N = 291), followed by animals (N = 72) and some strains were isolated from the environment (N = 6). One hundred and seventy strains were commensal whereas five and 187 were responsible of intestinal and extra-intestinal infections, respectively. The genomes of 295 strains were previously available, while the remaining 75 were sequenced as part of this work by Illumina technology as described previously [37]. The genome sequences of all strains are available through Figshare [76].

The construction of the *irp2* deletion mutants of the NILS9 and NILS46 strains was achieved following a strategy adapted from Datsenko and Wanner [77]. Primers used in the study are listed in S7 Table. In brief, primers used for gene disruption included 44–46 nucleotide homology extensions to the 5'- and 3' regions of the target gene, respectively, and additional 20 nucleotides of priming sequence for amplification of the resistance cassette on the template plasmids pKD4. The PCR product was then transformed into strains carrying the helper plasmid pKOBEG expressing the lambda red recombinase under control of an arabinose-inducible promoter [78]. Kanamycin resistant transformants were selected and further screened for correct integration of the resistance marker by PCR. For elimination of the antibiotic resistance gene, helper plasmid pCP20 was used according to the published protocol. PCR followed by Sanger sequencing of the mutants were performed to verify the deletion and the presence of the expected scar.

### Yersiniabactin detection assay

Production of the siderophore yersiniabactin was detected and quantified using a luciferase reporter assay as described elsewhere [18,79]. Briefly, bacterial strains were cultivated in NBD medium for 24 hours at 37˚C. Next, bacteria were pelleted by centrifugation and the supernatant was added to the indicator strain WR 1542 harbouring plasmid pACYC5.3L. All the genes necessary for yersiniabactin uptake are located on the plasmid pACYC5.3L, i.e. *irp6*, *irp7*, *irp8*, *fyuA*, *ybtA*. Furthermore, this plasmid is equipped with a fusion of the *fyuA* promoter region with the luciferase reporter gene. The amount of yersiniabactin can be quantified semi-quantitatively, as yersiniabactin-dependant upregulation of *fyuA* expression is determined by luciferase activity of the *fyuA-luc* reporter fusion.

### Mouse virulence assay

Ten female mice OF1 of 14–16 g (4 week-old) from Charles River (L'Arbresle, France) received a subcutaneous injection of 0.2 ml of bacterial suspension in the neck ($2 \cdot 10^{8}$ colony

forming unit). Time to death was recorded during the following 7 days. Mice surviving more than 7 days were considered cured and sacrificed[14]. In each experiment, the *E. coli* CFT073 strain was used as a positive control killing all the inoculated mice whereas the *E. coli* K-12 MG1655 strain was used as a negative control for which all the inoculated mice survive [27]. The data were available for 134 strains from our previous works whereas the remaining 236 strains were tested in this study (S1 Table). For the mutant assays, 20 mice per strain were used to obtain statistical relevant data. The data was analysed using the lifeline package v0.21.0 [80].

## Association analysis

All genome-wide association analysis were carried out using pyseer, version v1.3.4 [25]. All input genomes were re-annotated using prokka, version v1.14.5 [81], to ensure uniform gene calls and excluding contigs whose size was below 200 base pairs. The core genome phylogenetic tree was generated using ParSNP [82] to generate the core genome alignment and gubbins v2.3.5 [83] to generate the phylogenetic tree. The strain's pangenome was estimated using roary v3.13.0 [84]. Unitigs distributions from the input genome assemblies were computed using unitig-counter v1.0.5. The association between both unitigs and gene presence/absence patterns ("pangenome") and phenotype (expressed as number of mice killed post-infection) was carried out using the FastLMM [85] linear mixed-model implemented in pyseer, using a kinship matrix derived from the phylogenetic tree as population structure. For both association analysis we used the number of unique presence/absence patterns to derive an appropriate multiple-testing corrected p-value threshold for the association likelihood ratio test ($2.16E^{-08}$ and $5.45E^{-06}$ for the unitigs and pangenome analysis, respectively). Unitigs significantly associated with the phenotype were mapped back to each input genome using bwa mem v0.7.17-r1188 [86] and betools v.2.29.2 [87], using the pangenome analysis to collapse gene hits to individual groups of orthologs. A sample protein sequence for each groups of orthologs where at least one unitig with size 20 or higher was mapped was extracted giving priority to strain IAI39 when available, given it was the only strain with a complete genome available [88]; those sample sequences were used to search for homologs in the uniref50 database from uniprot [89] using blast v2.9.0 [90]. Each group of orthologs was then given a gene name using both available literature information and the results of the homology search. GO terms annotations were determined by submitting the protein sequence of each gene with associated unitigs to the eggnog-mapper website. GO terms enrichment was determined using goatools v1.0.6 [91]. Those genes with associated unitigs mapped to them and frequency in the pangenome > 0.9 were termed "core genes"; we searched for those genes in the E. coli K-12 genome (RefSeq: NC_000913.3) using blast v2.9.0 [90]

## Power simulations

Statistical power was estimated using a non-overlapping set of 548 complete *E. coli* genomes downloaded from NCBI RefSeq using ncbi-genome-download v0.2.9 on May 24th 2018. Each genome was subject to the same processing as the actual ones used in the real analysis (re-annotation, phylogenetic tree construction, pangenome estimation). The gene presence/absence patterns were used to run the simulations, in a similar way as described in the original SEER implementation [24]. Briefly, for each sample size, a random subset of strains was selected, and the likelihood ratio test p-value threshold was estimated by counting the number of unique gene presence/absence patterns in the reduced roary matrix. For each odds ratio tested, a binary case-control phenotype vector was simulated for the strains subset using the

following formulae:

$$P_{case \vee variant} = \frac{D_e}{MAF}$$

$$P_{case \vee novariant} = \frac{\frac{S_r}{S_r+1} - D_e}{1 - MAF}$$

Were $S_r$ is the ratio of case/controls (set at 1 in these simulations), $MAF$ as the minimum allele frequency of the target gene in the strains subset, and $D_e$ the number of cases. pyseer's LMM model was then applied to the actual presence/absence vector of the target gene and the likelihood ratio test p-value was compared with the empirical threshold, using the same population structure correction as the real analysis. The randomization was repeated 20 times for each gene and power was defined as the proportion of randomizations for each sample size and odds ratio whose p-value was below the threshold. To account for the influence of allele frequency on statistical power we picked 5 random genes for each allele frequency bin in the range [0.1–0.9].

## Correlations with growth profiles

The previously generated phenotypic data [28] for 186 strains over 370 total were used to compute correlations with both the number of mice killed after infection and presence/absence of the associated virulence factors. The data was downloaded from the ecoref website (https://evocellnet.github.io/ecoref/download/) and the pearson correlation with the s-scores (*i.e.* the normalized growth score for each strain in each condition [92]) was computed together with the correlation p-value. Prediction of tetracycline resistance was carried out using staramr v0.7.1 with the ResFinder database [93]. Four predictors, one for virulence (number of killed mice post-infection) and one for presence of the HPI, aerobactin and the *sitABCD* operon were built using the random forest classifier algorithm implemented in scikit-learn v.022.0 [94], using the s-scores as predictors. The input was column imputed, and 33% of the observations were kept as a test dataset, using a "stratified shuffle split" strategy. The remainder was used to train the classifier, using a grid search to select the number of trees and the maximum number of features used, through 10 rounds of stratified shuffle split with validation set size of 33% the training set and using the F1 measure as score. The performance of the classifiers on the test set were assessed by computing the area under the receiver operating characteristic curve (ROC-curve). For each predictor we derived the expected random baseline empirically by constructing a set of 15 predictors by shuffling the labels of the target vector, and keeping the training pipeline the same. We pooled the 15 random predictors and derived the average ROC and precision-recall curves with a 95% confidence interval.

## Software libraries

Code is mostly based on the Python programming language and the following libraries: numpy v1.17.3 [95], scipy v1.4.0 [96], biopython v1.75 [97,98], pandas v0.25.3 [99], pybedtools v0.8.0 [100], dendropy 4.4.0 [101], ete3 v3.1.1 [102], statsmodels v0.10.2 [103], matplotlib v3.1.2 [104], seaborn v0.9.0 [105], jupyterlab v1.2.4 [106] and snakemake v5.8.2 [107].

## Ethics statement

All animal experimentations were conducted following European (Directive 2010/63/EU on the protection of animals used for scientific purposes) and national recommendations (French

Ministry of Agriculture and French Veterinary Services, accreditation A 75-18-05). The protocol was approved by the Animal Welfare Committee of the Veterinary Faculty in Lugo, University of Santiago de Compostela (AE-LU-002/12/INV MED.02/OUTROS 04).

## Supporting information

**S1 Fig. Simulations of statistical power on a non-overlapping set of complete *E. coli* genomes, using the 5 random genes for each frequency bin, repeating the simulation 20 times for each gene and odds ratio.** The shaded area indicates the 95% confidence interval. The dotted red line indicates the sample size used in the actual analysis. AF, allele frequency.
(TIFF)

**S2 Fig. HPI structure conservation across strains.** One strain per phylogroup or species is shown, using the same color scheme as Fig 1E for each gene.
(TIFF)

**S3 Fig. Location of core genome genes with associated unitigs mapped to them (red) with respect to the High Pathogenicity Island (HPI, black).** The genome annotation of strain IAI39 is used as reference. Gene names were derived from *E. coli* K-12.
(TIFF)

**S4 Fig. Presence/absence patterns of known virulence factors.** Solid color indicates presence, light grey indicates absence. Phenotypes (number of killed mice) and phylogroup or species of each strain are reported as in Fig 1A. "Other virulence factors" are (from inside the ring towards the outside): *sfaD*, *sfaE*, *ompT*, *traT*, *hra2*, *papC*, *iha*, *ireA*, *neuC*, *hlyC*, *clbQ* and *cnf1*.
(TIFF)

**S5 Fig. Empirical random predictors for virulence and the presence of iron capture systems from high-throughput growth data.** Each line except the "Random predictor" represents the mean of 15 predictors built with suffled labels for the target variable. Vertical bars represent the 95% confidence interval.
(TIFF)

**S1 Table. Strains' information, including virulence phenotype.**
(XLSX)

**S2 Table. Summary of the 81 genes with at least one mapped unitig.**
(XLSX)

**S3 Table. GO terms enrichment analysis for the 81 genes with at least one mapped unitig.**
(XLSX)

**S4 Table. Survival analysis for NILS9 and NILS46 wild-type and HPI mutants.**
(XLSX)

**S5 Table. Correlation between growth on stress conditions (s-scores) and both virulence and presence of the HPI.**
(XLSX)

**S6 Table. Feature importance for each growth condition in the random forests predictor for virulence and HPI presence.**
(XLSX)

**S7 Table. List of PCR primers used in this study.**
(XLSX)

## Acknowledgments

We are grateful to Ivan Matic for discussion on the effect of indole.

## Author Contributions

**Conceptualization:** Marco Galardini, Pedro Beltrao, Erick Denamur.

**Data curation:** Olivier Clermont, Alexandra Baron.

**Formal analysis:** Marco Galardini.

**Funding acquisition:** Pedro Beltrao, Erick Denamur.

**Investigation:** Marco Galardini, Olivier Clermont, Alexandra Baron, Bede Busby, Sara Dion, Sören Schubert.

**Project administration:** Erick Denamur.

**Software:** Marco Galardini.

**Supervision:** Erick Denamur.

**Writing – original draft:** Marco Galardini, Erick Denamur.

**Writing – review & editing:** Marco Galardini, Pedro Beltrao, Erick Denamur.

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
