## [Decision Letter · Decision Letter 0]

9 Jun 2020

Dear Dr Galardini,

Thank you very much for submitting your Research Article entitled 'Major role of iron uptake systems in the intrinsic extra-intestinal virulence of the genus Escherichia revealed by a genome-wide association study' to PLOS Genetics. Your manuscript was fully evaluated at the editorial level and by three independent peer reviewers. The reviewers appreciated the attention to an important problem, but raised some substantial concerns about the current manuscript. Based on the reviews, we will not be able to accept this version of the manuscript, but we would be willing to review again a much-revised version. We cannot, of course, promise publication at that time.

If you decide to revise the manuscript for further consideration at PLOS Genetics, please aim to resubmit within the next 60 days, unless it will take extra time to address the concerns of the reviewers, in which case we would appreciate an expected resubmission date by email to plosgenetics@plos.org.

[LINK]

We are sorry that we cannot be more positive about your manuscript at this stage. Please do not hesitate to contact us if you have any concerns or questions.

Yours sincerely,

Xavier Didelot

Associate Editor

PLOS Genetics

Josep Casadesús

Section Editor: Prokaryotic Genetics

PLOS Genetics

Reviewer's Responses to Questions

**Comments to the Authors:**

Reviewer #1: In this manuscript, Galardini et al. describe a GWAS approach to understand the genetic underpinnings of a virulence phenotype in E. coli. Despite a modest sample size of genomes, they identify a top hit locus (HPI) with a large effect size, and a few other significant loci also involved in iron capture. They validate the HPI locus using knockout experiments in a mouse model of virulence, and also exploit an existing dataset of E. coli growth curves under different culture conditions to ask which growth conditions are correlated (or anti-correlated) with virulence, or the presence/absence of HPI or other top GWAS hits. This last analysis reveals a putative ‘collateral sensitivity’ in which virulence, HPI and other GWAS hits are anti-correlated with growth on sub-inhibitory concentrations of antibiotics in the presence of indole. Based on this result and prior experiment evidence, they conclude that iron may be deleterious to bacterial cells in the presence of antibiotics and other stressors. Thus, the presence of HPI and other iron capture systems may be necessary for virulence but deleterious in the presence of antibiotics. Overall, this study is a solid application of bacterial GWAS to an interesting phenotype (virulence in a mouse model). The synthesis with a larger phenotypic screening dataset is also appreciated, adding value to both studies. I enjoyed reading the paper, and suggest the following to improve it:

1) In my view, the idea of ‘collateral sensitivity’ between virulence factors and antibiotic resistance is the most valuable part of the study, and thus the most important part to get right. However, I had some trouble understanding some of these analyses (last results section) and what conclusions could be drawn. First, ‘collateral sensitivity’ is used in the abstract and throughout the manuscript, but never properly defined. The term is normally used to refer to selection for one antibiotic resistance resulting in increased sensitivity to a second antibiotic, but here it is used in a slightly different (but presumably analogous) context. It should be defined, with reference to earlier literature on this topic. Second, it is a bit unclear what can be concluded about the mechanisms of collateral sensitivity based on the phenotypic screen correlations and machine learning analyses. I appreciate that the conclusions are supported in a Discussion paragraph that brings in other experimental evidence (lines 272-291) but parts of the analyses could be better explained. For example, I also appreciated the attempt to determine what is driving the negative correlations between virulence, HPI and growth with indole + antibiotics (lines 190-203). The authors state that the Pearson’s correlation coefficients are ‘comparable’ (line 199), but it is not really clear if they are really of similar magnitude without confidence intervals. Ideally, I think that partial correlations should be computed to assess the correlation between HPI/aerobacterin/sitABCD and growth on tetracycline after controlling (or partialling out) the effect of the presence of known tetracycline resistance genes. Otherwise, the results are inconclusive.

2) The authors also chose to include several members of the genus in the GWAS analyses, in addition to just E. coli. However, it is not clear if the addition of these extra species help or hinder the analysis. From Fig S3, it seems that the long branches in the phylogeny might be adding more noise (e.g. bigger pan genome) but not adding to the association signal which seems to be coming mainly from E. coli. The rationale for including multiple species should at least be discussed.

3) The rationale for the experiments presented in the paragraph starting on line 133 is not clear. The paragraph begins by describing previous evidence that deletions in the HPI locus usually, but not always, reduce virulence. It is thus unclear why further experiments are necessary. The authors state the goal is to have a ‘broader view’ (line 139) but this is quite vague. Moreover, if the goal is to have a broader view (e.g. why is the attenuation phenotype observed when deletions are done in some lineages but not others?) this would probably require studying many more deletion mutations than just two.

4) Random forests analyses: To assess whether the trained models are significantly better than random, I would like to see the models trained multiple times on data with shuffled labels, to create a null distribution. This would allow to calculate a p-value for the ‘real’ model to be better (ie. better AUC or F1 score) than expected by chance. Currently, the line x=y plotted in fig 3E is not really informative. In reality, there will be a confidence interval around this line. Finally, if the aerobactin model is not reliable (as mentioned on line 209), perhaps it should be excluded from Figure 3G.

5) The article was generally well-written, but there are a few awkward phrases and a few non-standard idioms (e.g. the frequent use of the term ‘over’ when ‘out of’ is meant, as in a fraction or proportion).

Minor/specific comments:

- line 71: A brief definition and perhaps a citation or two would be useful for ‘collateral sensitivity.’

- paragraph starting on line 87 explaining the rationale for GWAS:

- line 89-91: large sample size will not necessarily break up the clonal frame. This depends entirely on whether additional samples bring greater phylogenetic diversity, and whether newly sampled phylogenetic groups have independent acquisitions of the genotype and phenotype of interest.

- line 94-7: unclear why the genetic determinants of virulence should have high penetrance.

-line 36: taxons should be taxa

-line 87: unitigs should be briefly defined here

- line 101: its should be it is

- line 107-108: unclear and awkward phrasing.

- line 113: how were other virulence factors defined? Presumably from the literature, but is this a comprehensive list of virulence factors?

- line 114-115: unclear if these unitigs pass the threshold or not

- line 115: It would be worth connecting these 33 genes to Fig 1D, presumably the points off the diagonal, with frequency close to 1?

- line 133: “KO” gene should be ‘Gene knockout”

- line 134: Please specify that ‘the studies’ refers to previous studies (E.g. ref 17)

- line 349: presumably contigs smaller than 200bp were excluded?

- line 354: It is implied that the pan genome association is between each coding gene and the phenotype, but this should be specified. Unitigs should also be defined.

- line 359: it is not clear if and how these p-value threshold are corrected for multiple hypothesis tests.

- line 366: where should be were

- lin 395: Please define s-scores.

Figure 1. In panel B, the points are sized according to the gene length fraction, but the significance of this is not made clear in the main text. I guess it shows that points with low p-values tend to be complete genes, whereas those with higher p-values tend to be gene fragments and thus possibly noise? Either this should be explained, or this detail removed from he plot. In panel C, it is unclear what distinguishes “other genes” in red from “all genes” in grey. Is it a p-value cutoff?

Fig S1: The pks2 gene was used as the target ’true positive,’ but it is not really clear how the precise gene identity would matter in these power simulations. Is it because the ‘real’ distribution of pks2 presence/absence across genomes was used, or simply the real fraction of genome containing the gene? If similar results were obtained for another gene (fabG), it is not clear how this should be interpreted. Is this gene present at lower frequency? Presumably the power depends partly on the frequency, but also on the extent to which genes are distributed in different clades of the phylogeny. These power calculations are certainly worthwhile, but deserve some more explanation.

It is also unclear if the simulation GWAS was corrected for population structure (the kinship matrix) as in the real GWAS?

Also, “unrelated” is probably not the right term here, because all E. coli are likely related (i.e. new lineages are not being sampled in each study). Perhaps “non-overlapping” sample of E. coli or something that effect would be more accurate.

Data availability: The genomes are all available in Figshare, which is a convenient way of accessing this particular combined dataset. However, I just want to make sure that any new genomes reported are also deposited in NCBI before publication.

Reviewer #2: General comments:

The authors report the results of a generally well-done, unbiased, WGS-based study designed to identify genetic and phenotypic correlates of lethality in a mouse sepsis model. Strengths include the comparatively large and diverse strain collection and the novel combination of WGS analysis, in vivo virulence assessments, and machine learning. Many opportunities exist to improve clarity, including regarding the rationale for some aspects of the study.

Specific comments:

1. Lines 28-29, the phrase "which hints at collateral sensitivities associated with intrinsic virulence" is unclear and confusing, in part because the preceding phrase mentions "antimicrobials". (A similar phrase in line 71 is also confusing.)

2. Line 35 would be clearer if "the genus" were used instead of "it", which is ambiguous.

3. Lines 79-80 would be clearer if the host species for the clinical and commensal isolates were to be specified. This appears in line 306-307, but is relevant here.

4. Line 87, the unfamiliar (to this reviewer) technical term "unitigs" should be explained at first usage.

5. Line 101 would be more grammatical if "if its" were to be replaced by "is", to match the "is" that appears earlier in the sentence; even better would be "instead is".

6. Line 102 seems to be missing a word, such as "analysis", after "association".

7. Line 103, the intended meaning of "showing" presumably is "which showed" or (as a new sentence) "This showed...".

8. Line 105, presumably "at least one" is meant, rather than "at least an".

9. Lines 107 and 108, the meaning of the phrase "it's the presence of these genes to be associated with virulence" is unclear.

10. Lines 108 and 109, this sentence refers to both genes/operons per se and their presence as being associated with virulence. Both are correct/acceptable, but the authors should go with one or the other, not use both in successive phrases. Consistent usage is needed here.

11. Lines 98-118, this paragraph is very long and contains several different ideas/topics. Ideally, it would be broken into smaller, more accessible, more thematically unified units.

12. Lines 113 and 114, it's confusing to have "known virulence factors" seemingly contrasted with the HPI and the aerobactin and sitABCD operons, since the latter also represent (or encode) known virulence factors. This point of confusion could be remedied by inserting "other" before "known" in line 113.

13. Line 115, "the remaining 33 genes" is confusing, because this number has not been mentioned up to now. Which 33 genes are these?

14. Line 118, which genes are "those genes" is unclear.

15. Fig. 1A, the lower key is labeled "Phylogroups", but the non-coli species and clades within the genus Escherichia aren't phylogroups, at least not in the usual sense of this term. Also, the red of the phylogroup ring blends with the red of the killer % ring, obscuring which is which. Furthermore, it's not immediately obvious which key applies to which ring; use of lines to connect each key with the corresponding ring would add clarity.

16. Fig. 1B and 1D, the meaning of the color code is unclear. Does the key in 1C apply here, too?

17. Fig 1C, no need to abbreviate OG (which is confusing); it's spelled out in 1D, so that could be done also in 1C. Also, the key is unclear: it shows red for "Other genes" (not further specified", and gray for "All genes", but "All" literally would subsume all the subcategories listed above in the key, including "Other genes".

18. Fig. 1D, AG is undefined. However, rather than defining it, a preferable approach would be to spell it out in the axis label.

19. Line 138, the term "this study" is ambiguous, since other studies were just cited; perhaps one of them is meant. Preferable wording would be "the present study".

20. Line 142, ST should be defined early on, at first mention.

21. Lines 135-143 implicitly suggest that previous experimental studies regarding the contribution of the HPI to virulence were limited to group B2 strains, and specifically to strains 536 and NU14. This, and the statement regarding achieving a broader phylogenetic assessment by studying strains from groups D (ST69) and A (ST10), overlooks a previous report of a virulence analysis, using the same mouse sepsis model, of an irp2 knockout of a strain from group D (ST69). One of the present authors coauthored that report (doi: 10.1016/j.micpath.2018.04.048.)

22. Line 146, no need to extend P values so many places beyond the decimal point. This implies false precision, and clutters the report without adding value.

23. Fig. 2A should have its axes flipped, i.e., to place strains [the independent variable] on the X-axis, and RLU [the dependent variable] on the Y-axis. However, RLU should be spelled out; there's no need to save space by abbreviating it, which interferes with comprehension. What 1e5 means is unclear.

24. Fig. 2B would work better if it were split into two figures, one for each strain and its mutant. The current combined graph is overly busy, and has some superimpositions that obscure the lines. There's no advantage (other than saving space) to including both strains in the same graph, because the two strains are not compared with each other; instead, each is compared with its own mutant, and combining all the data in one graph obscures these key comparisons.

25. Lines 168-169, the meaning of "186 over 370" is unclear. The same comment applies in several other locations.

26. Lines 169-171, this sentence is unsatisfyingly vague, with its qualifiers "relatively high" and "in certain conditions".

27. Line 176, the relevance of tobramycin is unclear. In general, more explanation is needed regarding why antibiotics and nutrient substrates were combined.

28. Line 180, to what "all growth conditions" refers is unclear, as is the intended meaning of "agree".

29. Lines 182-203, this is a long and dense paragraph. Accessibility would be improved by breaking it into smaller, more accessible units.

30. Line 182, in this sentence it's unclear what is correlated with growth on these various antibiotic-containing media. Also, this list of agents is unnecessarily complex, making it difficult to follow.

31. Lines 183 vs. 185, the implied distinction between "antibiotics" and "antimicrobial agents" is unclear. Ciprofloxacin, which strictly is regarded as an antimicrobial agent but not an antibiotic (because it's not a natural product), is classified here as an antibiotic, which is confusing.

32. Lines 199-200, the intended meaning of the complicated phrase "...which we found to be all comparable with the direct correlation between..." is unclear.

33. Figure 3 would be more accessible if each subfigure included an informative title, rather than just an alphabetic label that obliges the reader to consult the legend to learn what the data represent.

34. Line 233 and 287 (and elsewhere), the term "collateral sensitivities" is unclear.

35. Line 240, a word seems to be missing at the end of this sentence, after "in vivo".

36. Lined 240-242, it is unclear how the phylogenetic distribution of the HPI facilitates its detection by WGS. If the intended meaning is that WGS analysis facilitates assessment of the phylogenetic distribution of the HPI, that doesn't come across from the present wording.

37. Line 294, this statement seems overly broad, given that the only studied manifestation of extraintestinal virulence was lethality the mouse subcutaneous sepsis model. The findings might be specific to this model, or even this particular endpoint in this model.

38. Supplementary Figure 3 is very difficult to follow. Which rings show which variables, how the keys relate to the image, the basis for tree construction, and what point the image is intended to make are unclear.

39. The proposed hypotheses regarding mechanisms seem somewhat "after-the-fact". The rationale (if any) for studying the particular growth conditions that were selected is unclear.

Reviewer #3: E. coli is a gut commensal but also an intestinal and extra intestinal pathogen. This study by Galardini et al addresses determinants of extra intestinal virulence. First they apply a GWAS approach to find genes associated with increased virulence on a mix of 370 (326 E. coli ) commensal, pathogenic and environmental Escherichia strains. They identify the HPI (high pathogenicicty island), and iron uptake genes (aerobactin, sitABC) as associated to virulence. Next they validate their in silico results by in vivo deletion of HPI. Third, they seek to associate virulence to other known phenotypes, through data analysis and machine learning approach.

The results obtained in this study are solid and convincing.

This study validates GWAS as a powerful and unbiased approach to study important phenotypes such as virulence. Previous works using classical approaches have already identified HPI as a virulence factor (choosing to delete potential candidates and study of the phenotype). GWAS is an unbiased method allowing for the identification of such factors and candidates. However, the finding of the HPI is not the novelty here as this island was a known virulence factor. GWAS also identified iron uptake genes and the authors also mention 33 other genes (among which mtfA). It could be interesting to comment here about these other genes as well, as those could be newly identified virulence determinants (or candidates).

Page 6: In vivo deletion experiments were performed using a mouse model of sepsis, where authors have deleted irp2 and looked at attenuation of virulence. I’m assuming irp1 and irp2 belong to the HPI, no explanation is given about that.

There is a mix of different phypogroups and strains, and it’s sometimes difficult to follow throughout the paper. It could help if, every time, the authors could explain why they chose a given phyogroup/strain.

For example introduction and here: Lines 134-142: “The studies on the role of the HPI in experimental virulence gave contrasting results according to the strains’ genetic background. Among B2 phylogroup strains, HPI deletion in the 536 (ST127) strain did not have any effect in the mouse model of sepsis whereas this deletion in the NU14 (ST95) strain dramatically attenuated virulence. Two strains from this study belonging to B2 phylogroup/ST141 (IAI51 and IAI52) deleted in irp1 have attenuated virulence in the same model. To have a broader view of the role of the HPI in various genetic backgrounds, we constructed irp2 deletion gene mutants in two strains of phylogroup D (NILS46) and A (NILS9) belonging to STs (Sequence Types) frequently involved in human bacteraemia (ST69 and ST10, respectively).”

Results page 7: although this section is interesting, the link between (i) the GWAS approach to link virulence phenotype and virulence factors and (ii) high throughput study of phenotypic data to link HPI and other iron capture systems remains elusive for me. Maybe it’s just a matter of writing/explaining, but this sections appears very speculative, and the link made here between virulence and antibiotics seems indirect.

Line 169: “we observed a relatively high correlation”. Relatively?

Page 7, Lines 172 to the bottom of the page: difficult to read and should be rephrased more clearly. Line 176: The authors say here that there is a correlation between growth in bipyridyl and the presence of aerobactin and cite “bipyridyl+tobramycin”, why tobramycin?

Line 182: “we found strong positive correlations with growth on sub-inhibitory concentrations of several antibiotics”: positive correlation of what?

Page 8, line 206: “We used the commonly-used random forests machine l earning algorithm with appropriate partitioning of i nput data to tune hyperparameters and reduce overfitting”. Please explain.

Page 10: lines 272 and on: very speculative and I get the impression that several things are mixed up in this paragraph:

Line 276; Sub-lethal concentrations have been shown to induce indole production but at concentrations much lower (nM to µM) than the toxic concentration of indole (e.g. 5mM). I am not sure it would be accurate to make a parallel between the lower indole concentrations upon sublethal antibiotic treatment and the toxicity and ROS production by indole at mM concentrations..

Line 280: the authors are talking about indole related toxicity. Then the sentence beginning with “This toxicity has been shown to be partly iron mediated…(ref59)”. The toxicity of what exactly? Of indole? Of the antibiotic? Which antibiotic? Ref 59 cited here is not about indole but about trimetroprime, which is not mentioned by the authors. They rather say “tobramycin is the antibiotic involved in the negative correlation”. Correlation of what? Are they talking about their own data or another paper?

In the previous paragraph, the authors rather talk about aminoglycosides, tetracycline, rifampicin and amoxicillin, and resistance to these antibiotics in a fur mutant accumulating iron. On the other hand fur deletion also confers sensitivity to other antibiotics (which the authors do not mention but it’s also in ref 48 that is cited here). All these antibiotics have different modes of action and the effect of iron might be pleiotropic.

Other questions:

The study includes 370 among which 326 E. coli strains from 8 phylogroups, how/why were these strains chosen?

**Have all data underlying the figures and results presented in the manuscript been provided?**

Reviewer #1: Yes

Reviewer #2: Yes

Reviewer #3: Yes

PLOS authors have the option to publish the peer review history of their article (what does this mean?). If published, this will include your full peer review and any attached files.

Reviewer #1: No

Reviewer #2: No

Reviewer #3: No

---

## [Editor Report · Decision Letter 1]

20 Aug 2020

Dear Dr Galardini,

We are pleased to inform you that your manuscript entitled "Major role of iron uptake systems in the intrinsic extra-intestinal virulence of the genus Escherichia revealed by a genome-wide association study" has been editorially accepted for publication in PLOS Genetics. Congratulations!

Yours sincerely,

Xavier Didelot

Associate Editor

PLOS Genetics

Josep Casadesús

Section Editor: Prokaryotic Genetics

PLOS Genetics

Comments from the reviewers (if applicable):

**Data Deposition**

http://datadryad.org/submit?journalID=pgenetics&manu=PGENETICS-D-20-00657R1

**Press Queries**

---

## [Editor Report · Acceptance letter]

5 Oct 2020

PGENETICS-D-20-00657R1 

Major role of iron uptake systems in the intrinsic extra-intestinal virulence of the genus Escherichia revealed by a genome-wide association study 

Dear Dr Galardini, 

We are pleased to inform you that your manuscript entitled "Major role of iron uptake systems in the intrinsic extra-intestinal virulence of the genus Escherichia revealed by a genome-wide association study" has been formally accepted for publication in PLOS Genetics! Your manuscript is now with our production department and you will be notified of the publication date in due course.

With kind regards,

Matt Lyles

PLOS Genetics

On behalf of:
